# T-ILR: a Neurosymbolic Integration for LTLf

**Riccardo Andreoni**                                    RANDREONI@FBK.EU
*Fondazione Bruno Kessler, Trento, Italy*
*Free University of Bozen-Bolzano, Bozen-Bolzano, Italy*

**Andrei Buliga**                                         ABULIGA@FBK.EU
*Fondazione Bruno Kessler, Trento, Italy*
*Free University of Bozen-Bolzano, Bozen-Bolzano, Italy*

**Alessandro Daniele**                                    DANIELE@FBK.EU
*Fondazione Bruno Kessler, Trento, Italy*

**Chiara Ghidini**                                CHIARA.GHIDINI@UNIBZ.IT
*Free University of Bozen-Bolzano, Bozen-Bolzano, Italy*

**Marco Montali**                                MARCO.MONTALI@UNIBZ.IT
*Free University of Bozen-Bolzano, Bozen-Bolzano, Italy*

**Massimiliano Ronzani**                                  MRONZANI@FBK.EU
*Fondazione Bruno Kessler, Trento, Italy*

**Editors:** Leilani H. Gilpin, Eleonora Giunchiglia, Pascal Hitzler, and Emile van Krieken

## Abstract

State-of-the-art approaches for integrating symbolic knowledge with deep learning architectures have demonstrated promising results in static domains. However, methods to handle temporal logic specifications remain underexplored. The only existing approach relies on an explicit representation of a finite-state automaton corresponding to the temporal specification. Instead, we aim at proposing a neurosymbolic framework designed to incorporate temporal logic specifications, expressed in Linear Temporal Logic over finite traces ($\text{LTL}_f$), directly into deep learning architectures for sequence-based tasks.

We extend the Iterative Local Refinement (ILR) neurosymbolic algorithm, leveraging the recent introduction of fuzzy $\text{LTL}_f$ interpretations. We name this proposed method Temporal Iterative Local Refinement (T-ILR). We assess T-ILR on an existing benchmark for temporal neurosymbolic architectures, consisting of the classification of image sequences in the presence of temporal knowledge. The results demonstrate improved accuracy and computational efficiency compared to the state-of-the-art method.

## 1. Introduction

The integration of symbolic reasoning into deep learning (DL) architectures, commonly referred to as neurosymbolic (NeSy) learning, has emerged as a powerful paradigm to improve generalization capabilities in data-driven models. While a variety of NeSy frameworks have been proposed, the majority of these approaches primarily focus on first-order logics as the underlying symbolic formalism, only supporting reasoning over static domains (Badreddine et al., 2022; Daniele et al., 2023; Manhaeve et al., 2018). Nonetheless, the application of NeSy methods to dynamic, temporally structured environments, where knowledge is expressed in temporal logic, remains an open, underexplored problem.

Temporal logics, and especially Linear Temporal Logic, are employed across a range of domains—including formal methods (Pnueli, 1977), automated planning (De Giacomo et al., 2014), process mining (Di Ciccio and Montali, 2022), and reinforcement learning (De Giacomo et al., 2020). We believe that the exploration and adaptation of different NeSy approaches to temporal, dynamic domains can greatly benefit the research community.

Very little work exist on the combination of temporal logics and neural systems (see e.g., Badreddine et al. (2023); Garcez and Lamb (2003); Perotti et al. (2014); Di Francescomarino et al. (2017); Umili et al. (2023); Umili and Capobianco (2024)) and the list reduces even more if we focus on the direct, differentiable integration of temporal logic into NeSy architectures. The only notable approach to temporal NeSy integration relies on constructing finite-state automata that represent temporal formulae expressed in $\text{LTL}_f$ (Linear Temporal Logic on finite traces), which are subsequently used to define supervision signals over input sequences (Umili et al., 2023). While effective, this strategy may incur high computational costs due to the complexity of building finite-state automata (De Giacomo and Vardi, 2013).

Our work starts from the observation that the majority of recent NeSy approaches aim to directly embed the representation of the knowledge within the deep learning architectures (Badreddine et al., 2022; Daniele et al., 2023). Motivated by this observation we propose *Temporal Iterative Local Refinement (T-ILR)*, a novel framework for NeSy learning over sequential data with temporal knowledge expressed using $\text{LTL}_f$. Our approach builds upon the Iterative Local Refinement (ILR) (Daniele et al., 2023) algorithm that has shown excellent computational performances. We extend ILR to operate directly over $\text{LTL}_f$ specifications via their fuzzy interpretations. This allows for a differentiable, gradient-based optimization of temporal logic satisfaction within standard deep learning architectures. The proposed architecture includes a neural perception module that grounds the atomic propositions contained in the formula from raw observations, and a symbolic reasoning layer that encodes $\text{LTL}_f$ semantics through fuzzy temporal logics. Crucially, the combination of ILR with the fuzzy semantics of $\text{LTL}_f$ enables end-to-end differentiable training without the need for an external representation of the logical formulas (e.g., a finite-state-automata).

We empirically validate T-ILR by comparing it with the state of the art by expanding the benchmark for temporal symbol grounding in Umili et al. (2023) both in terms of sequences of increasing lengths and $\text{LTL}_f$ formulae built upon alphabets of increasing sizes. The results show a lead of T-ILR, both in terms of classification accuracy and computational efficiency.

The remainder of the paper is organized as follows: Section 2 reviews related work, Section 3 covers background concepts, and Section 4 introduces T-ILR. Finally, Sections 5 details and discusses the experimental evaluations and the results we have obtained.

## 2. Related Works

NeSy integration methods can be broadly categorized into two main approaches: those based on probabilistic logic and those based on fuzzy logic.

Probabilistic approaches typically rely on the optimization of Weighted Model Counting (WMC). Notable examples include DeepProbLog (Manhaeve et al., 2018), DeepStochLog (Winters et al., 2022), and Semantic Loss (Xu et al., 2018).

More closely related to our work are approaches grounded in fuzzy logic. These methods often fall into two categories: loss-based and model-based. *Loss-based* approaches inject

logical constraints into the training process by encoding them as a loss function or as a regularization term. This class includes Semantic Based Regularization (SBR) (Diligenti et al., 2017) and Logic Tensor Networks (LTN) (Badreddine et al., 2022). Despite their success, a key limitation of this class of methods is that logical constraints are typically not retained at inference time. In contrast, *model-based* approaches embed logical knowledge directly into the architecture, typically through additional layers or components that persist at inference time. Representative examples include KENN (Daniele and Serafini, 2019, 2022), C-HMCNN(h) (Giunchiglia and Lukasiewicz, 2021), and CCN+ (Giunchiglia et al., 2024). These methods enforce logical consistency by construction and are thus not subject to the same limitations as loss-based techniques.

Unlike loss-based methods, which have been shown to work better with product fuzzy logic (van Krieken et al., 2022), the model-based approaches are more commonly based on Gödel logic, as the logical layers require a closed-form formulation, which is not known for product logic, except for special cases (Daniele et al., 2023). Among this type of approaches, there is ILR (Daniele et al., 2023), upon which our work builds. It is a multi-layer refining algorithm that minimally modifies the neural network's output, making it consistent with the provided logical specifications (more details in Section 3.3).

All the approaches presented above are tailored to the integration of knowledge written using propositional or first order logics. When it comes to temporal logics, such as LTL and LTL$_f$ (see Section 3.1), the number of works available in the literature drops dramatically.

The direct, differentiable integration of temporal logic into NeSy architectures is recent, with Umili et al. (2023) providing the main example. Their work proposes a NeSy system for weakly-supervised learning, where input sequences are classified based on the satisfaction of a given LTL$_f$ formula. The formula is translated into a Deterministic Finite Automaton (DFA) and interpreted via fuzzy logic, resulting in a differentiable architecture that enables learning through back-propagation. While demonstrating the feasibility of integrating LTL$_f$ into DL pipelines, Umili et al. (2023) rely on DFA construction, which poses computational challenges due to the costly LTL$_f$ to DFA conversion (De Giacomo and Vardi, 2013).

We propose a differentiable, gradient-based optimization of LTL$_f$ formula satisfaction within a neural model, avoiding an external representation of the formula.

## 3. Background

We provide a brief introduction of Linear Temporal Logic on Finite Traces, and its fuzzy version, and the framework of Iterative Local Refinement (ILR).

### 3.1. Linear Temporal Logic on Finite Traces

Linear-time logics, that is, temporal logics predicating on traces, provide the most natural choice to express symbolic knowledge in our setting. Traditionally, traces are assumed to have an infinite length, as witnessed by Linear Temporal Logic (LTL) (Pnueli, 1977). In several application domains, such as planning and process mining, the dynamics of the system are more naturally captured using unbounded, but *finite*, traces. This led to LTL *on finite traces* (LTL$_f$) (De Giacomo and Vardi, 2013).

A LTL$_f$ formula $\varphi$ over a finite set $\mathcal{P}$ of propositional atoms follows the grammar:

$$\varphi ::= p \mid \neg\varphi \mid \varphi_1 \vee \varphi_2 \mid \mathsf{X}\varphi \mid \varphi_1 \mathsf{U} \varphi_2, \text{ where } p \in \mathcal{P}.$$

A *trace* $\tau$ over $\mathcal{P}$ is a finite, non-empty sequence $\tau = \langle \tau_1, \tau_2 \ldots, \tau_n \rangle$ where each $\tau_i, i > 0$ is a propositional assignment (in symbols $\tau_i \in 2^{\mathcal{P}}$) indicating which propositional atoms from $\mathcal{P}$ are true at instant $i$ in the trace. The length $n$ of $\tau$ is denoted $len(\tau)$. The grammar above extends propositional logic on $\mathcal{P}$ with formulae employing the temporal operators $\mathsf{X}$ and $\mathsf{U}$, representing *(strong) next* and *(strong) until*. Intuitively, when evaluated in an instant of the trace: $\mathsf{X}\varphi$ states that there exists a next instant, and $\varphi$ holds therein; $\varphi_1 \mathsf{U} \varphi_2$ states that $\varphi_2$ holds in the current or a later instant, and in all instants in between, $\varphi_1$ holds.

Formally, let $\varphi$, $\tau$, and $i$ be a LTL$_f$ formula, a trace, and an instant in the trace, respectively. We inductively define that $\varphi$ is true in instant $i$ of $\tau$, written $\tau, i \models \varphi$, as:

$\tau, i \models p$          if $p \in \tau_i$

$\tau, i \models \neg\varphi$        if $\tau, i \not\models \varphi$

$\tau, i \models \varphi_1 \vee \varphi_2$    if $\tau, i \models \varphi_1$ or $\tau, i \models \varphi_2$

$\tau, i \models \mathsf{X}\varphi$       if $i + 1 < len(\tau)$ and $\tau, i + 1 \models \varphi$

$\tau, i \models \varphi_1 \mathsf{U} \varphi_2$   if $\tau, j \models \varphi_2$ for some $j$ s.t. $i \leq j < len(\tau)$ and $\tau, k \models \varphi_1$ for every $k$ s.t. $i \leq k < j$

We say that $\tau$ satisfies $\varphi$, written $\tau \models \varphi$, if $\tau, 1 \models \varphi$.

The syntax and semantics of the other boolean connectives $\top, \bot, \wedge, \rightarrow$ are derived as usual. Further key temporal operators are derived from $\mathsf{X}$ and $\mathsf{U}$ as following: $\mathsf{R}$ (Release) $\varphi_1 \mathsf{R} \varphi_2 \equiv \neg(\neg\varphi_1 \mathsf{U} \neg\varphi_2)$; $\mathsf{G}$ (Globally) $\mathsf{G}\varphi \equiv \bot \mathsf{R} \varphi$ and $\mathsf{F}$ (Eventually) $\mathsf{F}\varphi \equiv \top \mathsf{U} \varphi$.

The process mining community has selected a number of patterns of LTL$_f$ formulas that are particularly significant for describing business processes in a declarative manner. These patterns constitute the DECLARE modelling language (Di Ciccio and Montali, 2022). An example of DECLARE pattern is the *chain response*, indicating that whenever activity $a$ occurs, $b$ must occur in the next time instant. This is formalised in LTL$_f$ as $\mathsf{G}(a \rightarrow \mathsf{X}b)$. Following Umili et al. (2023), we use DECLARE formulae in the evaluation of our approach.

### 3.2. Linear Temporal Logic on Finite Fuzzy Traces

In this work, we employ the *fuzzy interpretations* of LTL$_f$, as defined in Donadello et al. (2024), to enable its integration into gradient-based optimization via back-propagation. This logic, called FLTL$_f$, is the finite-trace counterpart of the infinite-trace temporal fuzzy logic FLTL from Lamine and Kabanza (2000); Frigeri et al. (2014).

Essentially, FLTL$_f$ expresses temporal properties of *fuzzy traces*. A fuzzy trace is a finite, non-empty sequence of functions $\langle \lambda_1, \lambda_2 \ldots, \lambda_n \rangle$, where for every $i \in \{1, \ldots, n\}$, function $\lambda_i$ assigns to each propositional symbol $p \in \mathcal{P}$ a corresponding real value $\lambda_i(p) \in [0, 1]$. Semantically, state formulas in FLTL$_f$ move from the crisp interpretation of LTL$_f$, to a fuzzy one where truth values of propositional formulas take on values in $[0, 1]$, using t-(co)norms to define the fuzzy truth values induced by conjunction, disjunction, and negation.

Temporal operators are interpreted as before, and thus yield fuzzy truth values that are defined as follows. The fuzzy truth value of $\mathsf{X}\varphi$ in instant $i$ is the fuzzy truth value of $\varphi$ evaluated in instant $i + 1$. The fuzzy truth value of $\varphi_1 \mathsf{U} \varphi_2$ is computed recursively using the equivalence $\varphi_1 \mathsf{U} \varphi_2 \equiv \varphi_2 \vee (\varphi_1 \wedge \mathsf{X}(\varphi_1 \mathsf{U} \varphi_2))$. The evaluation starts from the last instant

of the trace, where the formula reduces to $\varphi_2$, and proceeds backward through the trace. At each step, the conjunction and disjunction are computed using the chosen t-(co)norm, and the value of $\mathsf{X}(\varphi_1 \ \mathsf{U} \ \varphi_2)$ used in the recursion has already been computed at the previous recursion step, which corresponds to the next temporal instance in the trace.

As concrete instantiation for the t-(co)norm, FLTL$_f$ employs the Zadeh semantics (Donadello et al., 2024), where $\varphi \vee \psi$ yields the maximum among the truth values of $\varphi$ and $\psi$, $\neg \varphi$ yields 1 minus the truth value of $\varphi$, and the truth values for conjunction and implication are derived using the standard abbreviations. This choice is done for two reasons. First, it is compatible with that originally used in FLTL. Second, it retains the correspondence of derived temporal operators from LTL$_f$, where *release*, *globally*, and *eventually* are defined as syntactic sugar from $\mathsf{X}$ and $\mathsf{U}$. Under Zadeh, in FLTL$_f$ we get that the truth value obtained by natively interpreting the semantics of such derived temporal operators indeed coincides with the one obtained through the evaluation of their definition in terms of $\mathsf{X}$ and $\mathsf{U}$ only.

Finally, notice that the Zadeh semantics coincides with what is often called Gödel semantics, except that while Zadeh interpreted implication as material implication, under Gödel one can also opt for *residuum* as an alternative semantics (van Krieken et al., 2022).

### 3.3. Iterative Local Refinement (ILR)

Iterative Local Refinement (ILR) (Daniele et al., 2023) is a NeSy framework for enforcing logical constraints over the predictions of a neural network by iteratively refining its output. Differently from most NeSy approaches, which enforce logical constraints exclusively during training, ILR includes the knowledge directly into the model, allowing for imposing the constraints even during inference.

Let $\boldsymbol{\lambda} \in [0,1]^n$ denote the output of a neural network, and let $\varphi$ be a formula defined over $\boldsymbol{\lambda}$, interpreted under Gödel logic. Given a target truth value $t \in [0,1]$ (in this work, we always assume $t = 1$, enforcing full satisfaction of the constraints), the goal of ILR is to find a refined vector $\hat{\boldsymbol{\lambda}} = \boldsymbol{\lambda} + \boldsymbol{\delta}$ (with $\hat{\boldsymbol{\lambda}} \in [0,1]^n$) that satisfies $\varphi$ with fuzzy truth value $t$, while remaining close to the original prediction $\boldsymbol{\lambda}$. This is formalized as the following constrained optimization problem:

$$\mathrm{argmin}_{\boldsymbol{\delta}} \|\boldsymbol{\delta}\|_p \quad \text{s.t.} \quad \varphi(\boldsymbol{\lambda} + \boldsymbol{\delta}) = t, \quad 0 \le \boldsymbol{\lambda} + \boldsymbol{\delta} \le 1$$

where $\|\boldsymbol{\delta}\|_p$ represents the L$_p$ norm of the applied refinement, and $\varphi(\boldsymbol{\lambda} + \boldsymbol{\delta})$ represents the interpretation of $\varphi$ for the refined vector.

While the optimization problem is in general intractable, it can be solved analytically for formulas that involve a singular logical connective (conjunction, disjunction, implication and negation), providing a closed-form formula for simple logical constraints. Such formulas define the *minimal refinement functions* (MRF) for the various logical connectives. ILR finds an approximated solution to the original problem (involving a general formula $\varphi$) by decomposing $\varphi$ into a computational graph and applying the MRF to each node in a back-propagation like algorithm. Specifically, it consists of an iterative algorithm with two phases per iteration: *forward* (formula evaluation) and *backward* (formula refinement).

It is worth mentioning that the application of ILR corresponds to a new layer in the final architecture, which is then trained end-to-end as classical neural models. This can be done thanks to the ability of ILR to converge very fast to a (sub) optimal solution.

## 4. Temporal ILR (T-ILR)

We propose Temporal Iterative Local Refinement (T-ILR), a neurosymbolic framework designed to directly inject LTL$_f$ specifications into deep learning models for sequential tasks.

### 4.1. Problem Formulation

Consider a supervised learning task defined on temporal sequences of observations $x$. Let $x = \langle x_1, \ldots, x_n \rangle$ be an input sequence of length $n$, where each observation $x_i$ is defined in an arbitrary observation space $\mathcal{X}$ (e.g. images, sensor readings), that is $x_i \in \mathcal{X}$. Each sequence $x$ has assigned a label—usually a multiclass categorical variable—which can be represented as a finite set of propositional variables $\mathcal{Y} = \{y_1, \ldots, y_m\}$, and represents the objective of the learning task. In our setting, each observation sequence $x$ corresponds to a symbolic trace $\tau = \langle \tau_1, \ldots, \tau_n \rangle$ of the same length of $x$. In particular given a finite set of atomic propositions $\mathcal{P} = \{p_1, \ldots, p_{|\mathcal{P}|}\}$, which provides the symbolic representation of the observation space $\mathcal{X}$, each observation $x_i$ is mapped to the element $\tau_i \in 2^{\mathcal{P}}$.

Assume that knowledge is given as a set of LTL$_f$ formulas $\{\varphi_1, \ldots, \varphi_m\}$ defined over $\mathcal{P}$. Assume also that we can connect this knowledge with the labels in $\mathcal{Y}$, via a LTL$_f$ specification $\Phi$ defined over the set of propositional atoms $\mathcal{P} \cup \mathcal{Y}$, for example via the formula:

$$\Phi = (\varphi_1 \to y_1) \wedge (\varphi_2 \to y_2) \wedge \cdots \wedge (\varphi_m \to y_m) \tag{1}$$

Intuitively, $\varphi_1, \ldots, \varphi_m$ represent the body of knowledge used to assess the behaviour of the trace $\tau$, while Equation (1) connects this knowledge with the labelling of our observations. We enforce the logical constraints in $\Phi$ to solve the classification task $x \mapsto \mathcal{Y}$.

In this paper we focus on the version of this problem addressed by Umili et al. (2023), where the task consist of weakly supervised binary classification $\mathcal{Y} = \{y\}$ where the label $y$ directly corresponds to the satisfiability of an LTL$_f$ formula $\varphi$ by the traces $\tau$. In particular, a sequence of observations $x = \langle x_1, \ldots, x_n \rangle$ consists of digit images. A formula $\varphi$ expresses a temporal condition over the symbolic trace $\tau$, obtained by mapping each image in $x$ to its corresponding digit. For example, assume that the label consists in knowing whether our sequences satisfies the fact that an image of a 2 is always immediately followed by the image of a 0—in symbols $\mathsf{G}(2 \to \mathsf{X}0)$—or not, and that we represent the positive labelling with $y = \textit{true}$ and the negative labelling with $y = \textit{false}$, then Equation (1) becomes

$$\Phi = (\mathsf{G}(2 \to \mathsf{X}0) \to y) \wedge (\neg \mathsf{G}(2 \to \mathsf{X}0) \to \neg y).$$

### 4.2. Framework

The T-ILR framework (Figure 1) consists of two modules. The first is the neural, or perception, module, which maps each observation $x_i$ to a vector of fuzzy values. The second is the symbolic module, where a fuzzy counterpart $\tilde{\Phi}$ of the original LTL$_f$ formula $\Phi$ is evaluated and refined using the ILR layer (see Section 3.3). In the following, we provide a detailed explanation of each module of the T-ILR framework.

**The perception module.** This module consists of a parametric function $f_\theta : \mathcal{X} \to [0,1]^{|\mathcal{P}|}$, typically implemented as a neural network with learnable parameters $\theta$. This function maps each observation $x_i$ of the input sequence $x$ to a vector of fuzzy values

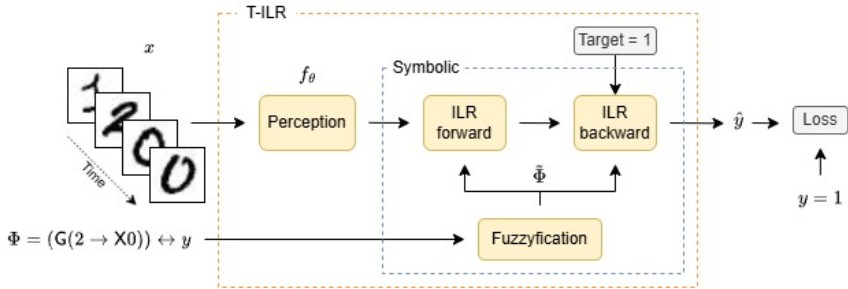

Figure 1: Overview of the T-ILR framework.

$f_\theta^j(x_i)$, $j = 1, \ldots, |\mathcal{P}|$, where each value represents the model's degree of confidence that the corresponding propositional atom in $\mathcal{P}$ holds for $x_i$.

Depending on the nature of the learning task, the output of the function $f_\theta(x_i)$ can be constrained accordingly. Under the mutual exclusivity (ME) assumption the outputs of $f_\theta(x_i)$ are defined using the softmax activation function, which forces their sum to one. Conversely, in the non-mutual exclusivity (NME) setting, the sigmoid activation is used.

**The symbolic module.** In order to integrate the LTL$_f$ specification $\Phi$ in a way that is compatible with back-propagation, we adopt the fuzzy semantics (FLTL$_f$) presented in Section 3.2. Specifically, the formula $\Phi$, which connects the trace $\tau$ to the propositional atoms defining the labels $\mathcal{Y}$, is transformed into its fuzzy counterpart $\tilde{\Phi}$, defined over the fuzzy trace $\lambda$ and the labels $\mathcal{Y}$.[1] While alternative t-norms are often favoured in gradient-based learning settings, our use of Gödel semantics preserves the intended logical properties of the FLTL$_f$ formalism, as defined in Donadello et al. (2024).

Given the observation sequence $x$, the output of the perception module is the sequence $\langle f_\theta(x_1), \ldots, f_\theta(x_n) \rangle \in [0,1]^{|\mathcal{P}| \times n}$, which can be interpreted as the fuzzy trace $\lambda$. Each element of this output corresponds to a value of the fuzzy trace evaluated at a specific propositional atom, namely $\lambda_i(p_j) := f_\theta^j(x_i)$. We initialise the label values $\mathcal{Y} = \{y_1, \ldots, y_m\} := \{0, \ldots, 0\}$, representing the targets over the observation sequence $x$.

Since the fuzzy formula $\tilde{\Phi}$ consists of a combination of individual logical connectives (conjunction, disjunction, negation, and implication), we can directly exploit an ILR layer as defined in Section 3.3. Specifically, we decompose $\tilde{\Phi}$ into a computational graph and apply the minimal refinement function (MRF) to each node corresponding to a single connective.

The ILR layer defined in this way computes an approximate solution to the minimal refinement of the fuzzy values required to satisfy the fuzzy formula ($\tilde{\Phi} = 1$). Specifically, in the forward step, it uses the values of the fuzzy trace $\lambda_i(p_j)$ and the target values $\mathcal{Y}$ to evaluate the satisfaction of the fuzzy formula. In the backward step, it produces the refinement values $\hat{\lambda}_i(p_j)$ and $\hat{y}_k$, for $k = 1, \ldots, m$.

The learning objective is to minimize a loss function $\mathcal{L}(\theta) = \sum_k \text{loss}(\hat{y}_k, y_k)$, such as cross-entropy, between the refined predicted satisfactions $\hat{y}_k$ and the ground truth labels $y_k$. Since all operations in the perception module $f_\theta$ and the symbolic module are differentiable, the gradient of $\mathcal{L}(\theta)$ with respect to $\theta$ can be computed, enabling back-propagation.

---

1. At this stage, the labels $\mathcal{Y}$ are treated as real-valued fuzzy interperations. A step function is applied at inference time to obtain a crisp label. This is similar to what happens in neural network training.

## 5. Evaluation

The evaluation we conducted is guided by the following research questions:

- **RQ1:** How does the proposed T-ILR framework compare to the state-of-the-art in neurosymbolic temporal reasoning?
- **RQ2:** What are the scalability limits of NeSy approaches for temporal reasoning in increasingly complex settings?

**RQ1** measures the benefits of T-ILR compared to the approach of Umili et al. (2023) in terms of performance (accuracy) in the setting proposed therein. **RQ2** measures the scalability of both approaches in settings with longer sequences and larger sets of symbols.

### 5.1. Baselines, Datasets, and Experimental Settings

We compare the T-ILR framework against the DFA approach by Umili et al. (2023), which achieves better performance compared to a pure deep learning-based approach.

The comparison is performed on the weakly supervised binary classification task introduced in Section 4.1, by leveraging the MNIST dataset (LeCun et al., 1998) to construct sequences of images. In this setting, each sample consists of a sequence of images paired with a binary label indicating whether the sequence satisfies or violates a given $\text{LTL}_f$ formula. This evaluates the neural perception module's ability to learn the underlying symbolic representation from visual data, based on the reasoning over their temporal properties, measured as the accuracy over the image classification task.

The evaluation protocol is split into two main parts. First, we answer **RQ1** by replicating the original evaluation setting of Umili et al. (2023) which uses a base of 20 $\text{LTL}_f$ formulas derived from DECLARE patterns, constructed with up to 2 propositional atoms. For each formula a dataset is created, composed of all the possible symbolic traces with length between 1 and 4, and labelled as accepted or rejected based on the satisfaction of the formula. Second, to systematically test scalability and answer **RQ2**, we introduce an extended protocol in which we vary the size of $\mathcal{P}$, ranging over $|\mathcal{P}| \in \{2, 3, 4\}$, and the maximum sequence length, ranging over $len(\tau) \in \{5, 10, 20\}$. This yields 9 possible combinations, for each 5 $\text{LTL}_f$ formulas $\varphi$ are sampled, resulting in 45 experimental configurations. The exponential growth of symbolic traces with $|\mathcal{P}|$ and $len(\tau)$ makes exhaustive dataset generation impractical, so we use stratified sampling to construct representative datasets. The details of the formula selection, dataset generation, and sampling procedures for all the experiments are provided in the supplementary material (Appendix A).

The experimental design involves the two scenarios introduced in Section 4 regarding the construction of the image sequences: mutual exclusivity (ME), and non-mutual exclusivity (NME). In the ME setting only one symbol holds at a timestep, while in the NME settings, at most two concurrent symbols can hold per timestep. Each training was run for 20 epochs, with a timeout of 60 minutes. Any experiment failing to complete within the limit was considered a timeout and not included in the final aggregated results. The T-ILR implementation and the experiments are available at github.com/andreoniriccardo/temporal_ILR.

### 5.2. Answering the Research Questions

Table 2: Average test accuracy (in %) and training runtime (in min). Superscripts denote the number of timed-out runs, which are counted as 60 minutes for averaging purposes.

| Setting | $|\mathcal{P}|$ | DFA Time | Sequence Length 5 | | | | Sequence Length 10 | | | | Sequence Length 20 | | | |
|---|---|---|---|---|---|---|---|---|---|---|---|---|---|---|
| | | | DFA | | T-ILR | | DFA | | T-ILR | | DFA | | T-ILR | |
| | | | Acc. | Time | Acc. | Time | Acc. | Time | Acc. | Time | Acc. | Time | Acc. | Time |
| ME | 2 | 0.01 | 89.73 | 4.05 | **100.00** | **0.77** | 70.83 | 6.21 | **99.91** | **0.91** | 69.63 | 9.87 | **99.84** | **1.39** |
| | 3 | 0.06 | 72.25 | 5.90 | **72.64** | **0.79** | 64.71 | 9.47 | **71.54** | **0.95** | 57.49 | 16.79 | **84.60** | **1.44** |
| | 4 | 10.68 | 40.56 | 29.28 | **49.52** | **0.89** | 34.39 | 40.30 | **66.85** | **1.36** | $25.31^{(2)}$ | $44.58^{(2)}$ | **60.47** | **4.32** |
| NME | 2 | 0.01 | 82.96 | 6.33 | **83.05** | **1.92** | **92.92** | 9.37 | 92.83 | **2.21** | 84.79 | 15.96 | **88.62** | **3.10** |
| | 3 | 0.09 | 65.51 | 8.67 | **72.00** | **2.16** | 66.59 | 12.45 | **72.08** | **2.57** | 68.12 | 23.53 | **74.65** | **3.67** |
| | 4 | 10.10 | $53.99^{(2)}$ | $38.21^{(2)}$ | **60.81** | **2.57** | $56.19^{(3)}$ | $43.61^{(3)}$ | **61.93** | **3.85** | $56.58^{(3)}$ | $47.45^{(3)}$ | **60.22** | **10.45** |

**Answering RQ1.** Table 1 summarizes the results of the overall accuracy averaged across the 20 LTL$_f$ formulas for DFA and T-ILR, following the evaluation protocol proposed by Umili et al. (2023). The results are split between the Mutually Exclusive (ME) and Non-Mutually Exclusive (NME) settings. In parenthesis, a number measuring how many times each methods outperforms the competitor across the 20 experiments.

Table 1: Test accuracy.

| Setting | DFA | T-ILR |
|---|---|---|
| ME | 84.12(6) | **87.94(9)** |
| NME | 76.83(7) | **83.70(13)** |

Overall, in the ME scenario T-ILR shows a moderate improvement in average accuracy compared to DFA. It outperforms the baseline in 9 out of 20 cases, while the DFA outperforms T-ILR in 6 cases. For the other 5 cases, the two methods perform similarly thus no significant differences are observed. In the NME scenario, where multiple atoms may co-occur, the advantage of T-ILR becomes more pronounced: it surpasses DFA in 13 out of 20 cases, while the DFA outperforms T-ILR the other 7 times. These results indicate that while T-ILR provides improvements in both settings, it particularly stands out in the more complex NME case, where the occurrence of concurrent events aligns well with the FLTL$_f$ formalism, which allows multiple propositional atoms to hold at a single timestep.

**Answering RQ2.** To answer **RQ2**, we leverage the results presented in Table 2. We highlight in parentheses the number of timeouts registered for the DFA method.

In the ME setting, T-ILR consistently outperforms DFA across all combinations of atoms number $|\mathcal{P}|$ and sequence lengths. The T-ILR method consistently maintains high accuracy, even as sequence length increases or more atoms are introduced—situations where DFA performance typically degrades significantly. For example, with $|\mathcal{P}| = 2$, T-ILR achieves perfect or near-perfect accuracy regardless of sequence length. Although accuracy naturally drops as $|\mathcal{P}|$ grows, T-ILR still retains a clear advantage over DFA. In the NME setting, T-ILR again demonstrates robust performance, outperforming DFA in 17 out of 18 configurations. While the gap between the two methods is slightly narrower than in the ME case, T-ILR still maintains a consistent advantage, particularly as $|\mathcal{P}|$ increases. Instead, by looking at Table 2 we observe large discrepancies between the execution times of the two methods, especially with longer sequences and bigger $|\mathcal{P}|$. In particular, we observe that in both the ME and NME setting, the runtime of the DFA method is always larger than that of T-ILR. This is especially true in the NME setting, with $|\mathcal{P}| = 4$, where the DFA method reaches

a timeout in 3 out of the 5 cases on different $\text{LTL}_f$ formulas. While DFA construction is done once per formula, parsing through the DFA during training to check the satisfiability of the network outputs is time-consuming, especially as the number of atoms and states grows. The performance decrease with a larger $|\mathcal{P}|$ is due to the learning problem becoming increasingly complex, as the number of symbolic sequences grows exponentially. This makes a single binary label increasingly insufficient to adequately supervise the grounding task.

## 5.3. Discussion

The results in Section 5.2 confirm T-ILR's effectiveness on temporal symbol grounding and validate the choice of integrating temporal logic directly via fuzzy semantics in a differentiable framework. Notably, T-ILR achieves superior accuracy, especially in more complex cases with longer sequences and larger alphabets. In contrast, the DFA-based method's performance degrades as task complexity grows, especially in the ME setting. These results suggest that T-ILR produces a signal that enables more stable, efficient learning within the perception module. This allows the model to better differentiate levels of fuzzy satisfaction of temporal formulae, maintaining robustness particularly in increasingly complex settings.

The evaluation in Section 5.2 also highlights the impact of the construction of the DFA structure and its use during the learning process on the execution time of the approach. This is due to the complexity and size of the DFA (De Giacomo and Vardi, 2013), which make both its construction and the repeated parsing during training costly, leading to exponential growth in execution time as sequence length increases. In contrast, T-ILR embeds the background knowledge directly within the neural model, limiting runtime growth.

These results have several implications for the NeSy community. First, they highlight the viability of fuzzy logic-based semantics as a bridge between symbolic and neural representations in the case of temporal domains. Second, the generalizability of T-ILR across both ME and NME scenarios underscores its potential for real-world applications, where the nature of symbol co-occurrence and sequence lengths are often unpredictable or variable. In this study we have leveraged the Gödel logic for the t-(co)norm semantics. However, other types of interpretations are available to model $\text{FLTL}_f$, such as the Łukasiewicz and Product t-(co)norms (Daniele et al., 2023). Finally, we believe that the semantics offered by $\text{FLTL}_f$ could be further extended and applied within other fuzzy logic-based NeSy frameworks to enrich the application of such approaches to these temporal domains.

## 6. Conclusions

In this work we propose T-ILR, a novel architecture based on the ILR algorithm that effectively integrates fuzzy temporal logic within a differentiable NeSy framework. Through the experimental evaluation, we demonstrate that the integration of T-ILR effectively outperforms existing NeSy methods for temporal domains in both accuracy and efficiency. The approach's flexibility across different scenarios and its grounding in fuzzy temporal semantics suggest promising directions for future research in NeSy temporal reasoning, where similar semantics could be used within other frameworks.

In the future, we would like to apply T-ILR to real-world domains like predictive and prescriptive process monitoring. Moreover, we aim to investigate how different semantics affect the fuzzy interpretation over temporal specifications in $\text{LTL}_f$.

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

## Appendix A. Supplementary Material

### A.1. Extended Evaluation Protocol

To systematically evaluate scalability, we designed an extended protocol by varying the number of symbols $p$, ranging over the values $|\mathcal{P}| \in \{2, 3, 4\}$, and the maximum length of the sequences $\tau$, with possible values $len(\tau) \in \{5, 10, 20\}$. This results in 9 distinct experimental combinations. For each possible value of $|\mathcal{P}|$, we sampled 5 LTL$_f$ formulas $\varphi$, resulting in 45 distinct configurations.

For $|\mathcal{P}| = 2$, we randomly sampled 5 formulas from the original pool. For more than 2 symbols we employ a conjunction of multiple formulas $\varphi_i$ with 2 symbols at a time. For example, for $|\mathcal{P}| = 3$ an LTL$_f$ formula $\varphi$ defined over symbols $p_1$, $p_2$, $p_3$ involves the conjunction of two formulas $\varphi_1$ (defined over symbols $p_1$, $p_2$) and $\varphi_1$ (defined over symbols $p_2$, $p_3$):

$$\varphi(p_1, p_2, p_3) = \varphi_1(p_1, p_2) \wedge \varphi_2(p_2, p_3)$$

As in the case of $|\mathcal{P}| = 2$, formulas $\varphi_1, \varphi_2$ were sampled from the original pool.

Compared to the original evaluation protocol of Umili et al. (2023), the increase in symbols and sequence length makes exhaustive use of all symbolic sequences impractical. For this reason, in the ME experiments, datasets are generated using stratified sampling: 1,000 symbolic sequences (from length 2 onwards) are labelled based on LTL$_f$ satisfaction, split into 500 training and 500 test sequences, and each symbolic sequence is converted into 5 MNIST image sequences (totaling 2,500 for training and testing each). Moreover, in the NME experiments, while dataset size remains the same, the number of possible sequences exponentially increases due to the overlap of symbols. To address this, all sequences up to length 4 are generated as in Umili et al. (2023), with 20% used for training. The remaining 80% is sampled up to the max length, keeping 2,500 sequences for training and testing.

Following the original protocol, each symbolic sequence was converted into a sequence of MNIST images. To maintain the integrity of the evaluation and avoid overfitting, images used to create training sequences were sampled from a pool kept entirely separate from the one used for test sequences.

For both the ME and NME settings, we employed the ADAM optimizer with a learning rate of 0.001 and a batch size of 64.

