# OpenReview forum: "T-ILR: a Neurosymbolic Integration for LTLf"
_nesyconf.org/NeSy/2025/Conference_Phase_2 — NeSy 2025 - Phase 2 Poster_

### Official Review · Reviewer_uWa1 · 2025-07-07
**Extending fuzzy-logic based neurosymbolic approaches to temporal logic is highly relevant and timely.**

**Rating:** 7
**Confidence:** 4

**Review:**

A proposal to extend fuzzy-logic based neurosymbolic frameworks such as LTN to linear temporal logic using the iterative local refinement approach.

This line of work is highly relevant to the research efforts in neurosymbolic integration, in particular in what concerns the representation capabilities of neural networks, as stated in the paper "without the need of an external representation of the logical formulas".

To help position the work in the broader context of temporal logic, it's worth noting (although some of the work below may not seem to be directly related to the proposed approach):

Work on Interval LTN (arguably more closely related to the paper than the ProbLog-based approaches): https://arxiv.org/abs/2303.17892

Encoding of temporal logic programming in the architecture of neural networks (as opposed to the loss function):

https://proceedings.neurips.cc/paper/2003/hash/347665597cbfaef834886adbb848011f-Abstract.html

Related application to runtime verification:

https://scholar.google.com/citations?view_op=view_citation&hl=en&user=Aef-vmEAAAAJ&cstart=20&pagesize=80&citation_for_view=Aef-vmEAAAAJ:YsMSGLbcyi4C

Early work about the extraction of automata from recurrent networks:

https://dl.acm.org/doi/abs/10.1162/0899766053630350

Although evaluation is limited to a few predicates and short sequences on the MNST dataset alone (which seems like an odd choice; there are better sequence learning and reasoning tasks to consider, not necessarily requiring a perception step), the results indicate scalability and an improvement compared with DFA.

Scalability is highlighted nicely in Table 3.

Business process modelling is a relevant and promising area of application for further evaluation.

Why is performance dropping considerably in Table 2 with the number of predicates?

In Table 1, what are the numbers inside brackets?

**Anonymity:**

Remain anonymous

---

### Official Review · Reviewer_j2Mt · 2025-07-08
**Paper breaks new ground, evaluation is still toy. Question about the choice of the Godel norm.**

**Rating:** 8
**Confidence:** 3

**Review:**

The paper addresses an understudied challenge, namely the integration of temporal logic in NeSy systems. The approach taken is to take a well understood and frequently used a-temporal approach (LTN), and then inject an existing temporal formalism LTL_f into the LTN framework. For this, LTL_f needs to be given a fuzzy interpretation, which is done in a straightforward way. The resulting system T-ILR is evaluated against the only other known temporal NeSy framework in a toy evaluation setting.

The work is definitely innovative and interesting, and opens new ground. The presentation is reasonable, although it takes 5 pages to present all the existing ingredients, which means that the actual description of T-ILR has to be squeezed into 2 pages, which have become very dense to read. The evaluation is in a rather toy setting, so more work remains to be done there to move this to a setting of realistic time series of input data and constraints, but it is sufficient to show the plausibility of the added value of T-ILR.

A technical question is the choice for the Godel-norm, which is known to be poor for learning (the cited Van Krieken paper).

Summarising, the paper breaks new ground, is reasonably presented (although too dense in the core pages), and does a preliminary evaluation (although a bit of a toy setting). I would like to see a motivation for the choice for the Godel-norm.

**Anonymity:**

Disclose identity

---

### Official Review · Reviewer_9VCw · 2025-07-09
**T-ILR: a Neurosymbolic Integration for LTLf**

**Rating:** 6
**Confidence:** 3

**Review:**

The paper presents a framework for the integration of temporal symbolic knowledge with Deep Learning architectures, leveraging the Iterative Local Refinement algorithm and the fuzzy interpretation of Linear Temporal Logic on finite traces. The framework, that does not exploit an external representation of the symbolic knowledge, is then compared with existing approaches based on the DFA translation of $LTL_f$ formulas.

The paper is in general well written, and the importance of addressing temporal reasoning in neurosymbolic AI is clearly stated. The topic is surely of interest, and the ability of managing temporal constraints without relying on external representations is an advantage of the proposed method w.r.t. other existing methods.

However, I have some concerns about the paper. The main contribution is a refinement of an existing algorithm, but it is not clear if and how the algorithm is modified or improved. Using it as part of a new framework could be fine, if the theoretical part was discussed in more details, and more experiments were provided. Nevertheless, the underlying theory is not discussed in general terms, but mainly by mean of examples (see e.g. Section 4.1). The provided experiments appear to me quite simple, and despite good results achieved in comparison with DFA based methods, I would like to see an example of the broader applicability of the framework. I think that these aspects should be addressed in more detail, in order to present more structured and principled results.

Pros:
- the paper is in general well written and easy to understand;
- addressing temporal constraints without relying on external structures is a clear advantage w.r.t. other existing methods;
- performance on the proposed experiments shows a significant improvement.

Cons:
- the underlying theory should be discussed in more detail: it is not very clear why and how it works;
- the proposed experiments are quite simple: due to the nature of the paper, I would expect at least one example of applicability on real world scenarios.

**Anonymity:**

Remain anonymous

---

### Official Review · Reviewer_TPdE · 2025-07-09
**The paper describes a neuro-symbolic architecture to tackle problems with temporal information. The experimental analysis is not convincing since the proposed method is only compared to another neuro-symbolic approach.**

**Rating:** 4
**Confidence:** 3

**Review:**

The theoretical part of the paper is clear and original, showing a method to embed linear temporal logic into machine learning. However, the experimental analysis is not convincing, since the proposed method is only compared to another neuro-symbolic approach (DFA).

**Anonymity:**

Remain anonymous